# Prospective, Multicenter Phase II Trial of Non-Pegylated Liposomal Doxorubicin Combined with Ifosfamide in First-Line Treatment of Advanced/Metastatic Soft Tissue Sarcomas

**DOI:** 10.3390/cancers15205036

**Published:** 2023-10-18

**Authors:** Angela Buonadonna, Simona Scalone, Davide Lombardi, Arianna Fumagalli, Alessandra Guglielmi, Chiara Lestuzzi, Jerry Polesel, Vincenzo Canzonieri, Stefano Lamon, Petros Giovanis, Sara Gagno, Giuseppe Corona, Maurizio Mascarin, Claudio Belluco, Antonino De Paoli, Gianpiero Fasola, Fabio Puglisi, Gianmaria Miolo

**Affiliations:** 1Department of Medical Oncology, Unit of Medical Oncology and Cancer Prevention, Centro di RiferimentoOncologico di Aviano (CRO), IRCCS, 33081 Aviano, Italy; abuonadonna@cro.it (A.B.); sscalone@cro.it (S.S.); dlombardi@cro.it (D.L.); arianna.fumagalli@cro.it (A.F.); or fabio.puglisi@uniud.it (F.P.); 2Oncology Department, Azienda Sanitaria Universitaria Giuliano-Isontina (ASUGI), 34100 Trieste, Italy; alessandra.guglielmi@asugi.sanita.fvg.it; 3Department of Cardiology, Azienda Sanitaria Friuli Occidentale (ASFO), Cardiology Unit at Centro di Riferimento Oncologico di Aviano (CRO), IRCCS, 33081 Aviano, Italy; chiara.lestuzzi@gmail.com; 4Unit of Cancer Epidemiology, Centro di Riferimento Oncologico di Aviano (CRO), IRCCS, 33081 Aviano, Italy; polesel@cro.it; 5Pathology Unit, Centro di Riferimento Oncologico di Aviano (CRO), IRCCS, 33081 Aviano, Italy; vcanzonieri@cro.it; 6Department of Medical, Surgical and Health Sciences, University of Trieste, 34129 Trieste, Italy; 7Unit of Oncology, Oderzo Hospital, Azienda ULSS 2 Marca Trevigiana, 31046 Oderzo, Italy; oncologiaoderzo@aulss2.veneto.it; 8Department of Oncology, Unit of Oncology, Santa Maria del Prato Hospital, Azienda ULSS 1 Dolomiti, 32032 Feltre, Italy; oncologia.fe@aulss1.veneto.it; 9Experimental and Clinical Pharmacology Unit, Centro di Riferimento Oncologico di Aviano (CRO), IRCCS, 33081 Aviano, Italy; sgagno@cro.it; 10Immunopathology and Cancer Biomarkers, Centro di Riferimento Oncologico di Aviano (CRO), IRCCS, 33081 Aviano, Italy; giuseppe.corona@cro.it; 11AYA Oncology and Pediatric Radiotherapy Unit, Centro di Riferimento Oncologico di Aviano (CRO), IRCCS, 33081 Aviano, Italy; mascarin@cro.it; 12Department of Surgical Oncology, Centro di Riferimento Oncologico di Aviano (CRO), IRCCS, 33081 Aviano, Italy; cbelluco@cro.it; 13Radiation Oncology Department, Centro di Riferimento Oncologico di Aviano (CRO), IRCCS, 33081 Aviano, Italy; adepaoli@cro.it; 14Department of Oncology, Azienda Sanitaria Universitaria Friuli Centrale (ASUFC), 33100 Udine, Italy; gianpiero.fasola@asufc.sanita.fvg.it; 15Department of Medicine, University of Udine, 33100 Udine, Italy

**Keywords:** soft tissue sarcoma, non-pegylated doxorubicin, doxorubicin, ifosfamide

## Abstract

**Simple Summary:**

This clinical investigation reports the results of a prospective, multicenter phase II trial designed to evaluate the activity and safety of combining non-pegylated liposomal doxorubicin (NPLD) with ifosfamide as a first-line treatment for advanced/metastatic STS. The study results demonstrate a remarkable overall response rate (ORR) alongside a satisfactory disease control rate (DCR) while maintaining acceptable levels of toxicity. The addition of NPLD to ifosfamide, showing high activity and a low toxicity profile, makes this drug combination a useful option for patients with advanced/metastatic STS. These findings provide a promising basis for further comprehensive research into the clinical application of this drug combination in this disease setting.

**Abstract:**

Doxorubicin is a widely used anticancer agent as a first-line treatment for various tumor types, including sarcomas. Its use is hampered by adverse events, among which is the risk of dose dependence. The potential cardiotoxicity, which increases with higher doses, poses a significant challenge to its safe and effective application. To try to overcome these undesired effects, encapsulation of doxorubicin in liposomes has been proposed. Caelyx and Myocet are different formulations of pegylated (PLD) and non-pegylated liposomal doxorubicin (NPLD), respectively. Both PLD and NPLD have shown similar activity compared with free drugs but with reduced cardiotoxicity. While the hand–foot syndrome exhibits a high occurrence among patients treated with PLD, its frequency is notably reduced in those receiving NPLD. In this prospective, multicenter, one-stage, single-arm phase II trial, we assessed the combination of NPLD and ifosfamide as first-line treatment for advanced/metastatic soft tissue sarcoma (STS). Patients received six cycles of NPLD (50 mg/m^2^) on day 1 along with ifosfamide (3000 mg/m^2^ on days 1, 2, and 3 with equidose MESNA) administered every 3 weeks. The overall response rate, yielding 40% (95% CI: 0.29–0.51), resulted in statistical significance; the disease control rate stood at 81% (95% CI: 0.73—0.90), while only 16% (95% CI: 0.08–0.24) of patients experienced a progressive disease. These findings indicate that the combination of NPLD and ifosfamide yields a statistically significant response rate in advanced/metastatic STS with limited toxicity.

## 1. Introduction

The anthracycline doxorubicin is one of the most widely used chemotherapeutic agents for the treatment of many cancers and is used as a first-line treatment alone or in combination in different tumor types, including soft tissue sarcomas (STS) [1,2,3,4]. The potential of doxorubicin is strongly limited by its adverse events, particularly, but not only, cardiotoxicity that limits the overall cumulative dose of the drug [5,6]. In addition, the pharmacokinetics/pharmacodynamic profile of the drug, with a very short half-life, represents another unfavorable factor. In the past, several analogs of doxorubicin have been designed and synthesized with the aim of maintaining the same activity with a lower toxicity profile. Epirubicin, also known as 4′-epidoxorubicin, was one of the earliest analogs of doxorubicin to be investigated, showing an equivalent activity and a more favorable toxicity profile when compared with doxorubicin in advanced STS [7]. Further investigations have demonstrated a dose-response relationship, indicating an increased response rate when epirubicin is administered at intensified doses in combination with ifosfamide and granulocyte-colony-stimulating factor (G-CSF) support [8,9]. Currently, the combination of epirubicin and ifosfamide still remains in Europe as one of the preferred treatment regimens for neoadjuvant or adjuvant chemotherapy in patients with high-risk STS [10]. Nevertheless, the true advantage of epirubicin over doxorubicin is still a subject of debate, and recent experiences have reported similar levels of cardiac toxicity between the two drugs. In order to improve the pharmacokinetics profile, innovative formulations of doxorubicin, which involve its liposomal encapsulation, have been introduced in clinical settings. They consist of two major distinct formulations based on pegylated liposomal (PLD: Caelyx™ or Doxil™) and non-pegylated liposomal doxorubicin (NPLD: Myocet™) [11,12,13,14].

PLD has been approved as a monotherapy in the US and Europe for the treatment of breast cancer of patients at increased cardiac risk, in advanced ovarian cancer, and AIDS-related Kaposi’s sarcoma, with the evidence of reduced cardiac toxicity compared with free doxorubicin, a prolonged serum half-life, with a similar efficacy reported in several clinical trials [11,15,16,17].

Interestingly, a recent Phase II study designed to evaluate the activity and safety of the combination of PLD with ifosfamide as a first-line treatment for patients with metastatic or locally advanced STS demonstrated an overall response rate (ORR) of 26.1% and a disease control rate (DCR) of 81.2% [18]. The undesired effect reported with the use of Caelyx is the Palmar–Plantar hand Erythrodysesthesia, also known as hand–foot syndrome (HFS), which can impair the quality of life of patients receiving the formulation and represents its dose-limiting toxicity. NPLD, while demonstrating similar efficacy to free doxorubicin and PLD, offers the significant advantage of lower toxicity in general, including a reduced incidence of HFS compared to both doxorubicin and PLD [19,20,21], which are approved treatments of metastatic breast cancer [11,22].

STS are a rare and heterogeneous group of mesenchymal tumors that account for approximately 1% of all adult tumors [3]. While surgery represents the main standard when the tumor is localized, for inoperable or advanced/metastatic disease, the standard chemotherapy in the first-line consists of doxorubicin alone or in combination with ifosfamide [23]. Although doxorubicin remains to date one of the most effective drugs for these tumors, objective responses are seen in less than 20% of patients, and 5-year survival for advanced STS remains low, being only 5% [24].

Here, we present the results of a multicenter, phase II trial combining NPLD and ifosfamide in advanced/metastatic STS.

## 2. Materials and Methods

### 2.1. Patients’ Characteristics

The study is a prospective phase II, one-stage, single-arm, multicenter open-label trial. The patients to be enrolled had to have a histological diagnosis of advanced/metastatic STS, including the histotypes liposarcoma, synovial sarcoma, fibrosarcoma, leiomyosarcoma, malignant fibrous histiocytoma, angiosarcoma (excluding fibrous solitary tumor and hemangioendothelioma), neurogenic sarcoma, sarcoma NOS, and others (clear cell sarcoma, epithelioid sarcoma, chondrosarcoma, glomangiosarcoma, PNET). Furthermore, additional inclusion criteria were: STS advanced for metastatic lesions or locally advanced not amenable to surgery; STS with progressive disease after 6 months from the end of adjuvant or neoadjuvant previous chemotherapy; age 18 years; Performance Status (according to Eastern Cooperative Oncology Group—ECOG) ≤ 2; life expectancy ≥ 3 months; measurable disease at least in one dimension; adequate bone marrow function (neutrophils ≥ 1.5 × 10^9^/L, platelets ≥ 100 × 10^9^/L); adequate renal function with Clearance of creatinine calculated by Cockcroft equation ≥ 60 mL/min or blood creatinine ≤ 1.6 mg/dl; adequate liver function with total bilirubin ≤ 1.5 mg/dl, Alkaline Phosphatase (ALP), Alanine Aminotransferase (ALT), and Aspartate Aminotransferase (AST) ≤ 2.5 times the upper normal limit (if liver metastases were present, the latter cut-off could be considered up to 5 times); normal Left Ventricular Ejection Function (LVEF), at least ≥ 50% evaluated with echocardiography; previous radiotherapy performed more than 4 weeks (8 weeks if extensive fields); written Informed Consent. The criteria for exclusion were: previous chemotherapy with cumulative doses of doxorubicin more than 300 mg/m^2^ or epirubicin more than 600 mg/m^2^; therapy within the previous 4 weeks with the drugs that are objects of this study; pregnancy or breastfeeding; fertile patients not taking adequate contraceptive therapy; concomitant uncontrolled infections; previous or active neoplasms (with the exception of carcinoma in situ of the uterine cervix and squamocellular carcinoma of the skin); clinical signs of brain and/or meningeal metastases (with the exception of pre-treated lesions now clinically stable); inadequate compliance.

The study was conducted under ethical principles, according to the Declaration of Helsinki and Good Clinical Practice (GCP) guidelines, after the submission and the consequent approval from the local Independent Ethics Committee (CEI).

Before the start of the study, written information and Informed Consent were provided to the subjects and/or their legal representatives.

### 2.2. Treatment and Study Design

The experimental treatment included 6 cycles of chemotherapy with NPLD (“Myocet” 50 mg/m^2^ on day 1 administered by a slow intravenous infusion of at least 60 min) plus ifosfamide (3000 mg/sqm on day 1, 2, and 3 with equidose MESNA) every 3 weeks with the first instrumental evaluation scheduled after the second cycle. Only patients with stable disease or partial/complete response after the first instrumental evaluation continued the treatment until the subsequent evaluations scheduled after the fourth and sixth cycles of chemotherapy or until disease progression or unacceptable toxicity. Instead, patients who experienced clinical deterioration within 3 weeks from starting treatment were deemed not evaluable and excluded from the study.

Concurrent radiation therapy was allowed to control bone pain or other symptoms. Antiemetic prophylaxis was recommended, according to practical guidelines.

Granulocyte Colony-Stimulating Factor (G-CSF) was administered, following each cycle, daily starting 7 days after day 1 of chemotherapy for 6 consecutive doses. Clinical tolerability to treatment was assessed by recording patient-reported or investigator-observed adverse events according to the Common Terminology Criteria for Adverse Events (CTCAE) version 3.0.

The primary endpoint was ORR, defined as the proportion of patients with confirmed partial or complete responses (PR and CR, respectively) at radiological assessment, using RECIST criteria version 1.0. Lesions were considered evaluable if at least one dimension was greater than 20 mm; a maximum of 8 lesions could be considered. In the case of a single lesion, histological confirmation was required. Response was assessed after the second, the fourth, and the sixth cycle.

Secondary endpoints were Overall Survival (OS), Progression-Free Survival (PFS), Duration of Response (DoR), and toxicity.

After completing the 6 cycles of treatment, patients were initiated into a follow-up program with clinical and instrumental assessments at 4 months for the first two years, at 6 months for the subsequent three years, and annually starting from the fifth year.

### 2.3. Statistical Analysis

ORR was calculated as the proportion of patients with confirmed partial or complete responses on the total patients, with corresponding 95% confidence intervals (CI) according to the Clopper–Pearson method.

For each patient, the time at risk was calculated from the date of first administration of NPLD plus ifosfamide to the date of death, recurrence, or last available follow-up, whichever occurred first. Survival probabilities for OS and PFS, with corresponding 95% CI, were estimated according to the Kaplan–Meier method [25]. Exploratory data analysis for the evaluation of factors associated with PFS and OS was performed with a Log-rank test. The hazard ratio (HR) of death, with corresponding 95% CI, was calculated by means of univariate and multivariable analyses according to the Cox proportional hazards regression model. The frequency and percentage of the maximum degree of each type of toxicity found were reported and evaluated according to CTCAE 3.0. Significance was claimed for *p* < 0.05 (two-sided). Statistical analyses were performed with R-4.3.1. software.

The Fleming one-stage design was the statistical methodology used to determine the appropriate sample size. In this specific design, we aim to detect a 15% difference in response rates [from 25% (po) to 40% (p1)] with a significance level [alpha, (α) of 0.05 and a type II error rate beta (β)] of 0.10%. The enrollment was set up to 80 patients, which represents the number needed to verify the hypothesis.

## 3. Results

Eighty patients with advanced/metastatic STS were enrolled in this multicenter study, conducted in hospitals located in the northeastern region of Italy. The clinical characteristics of the patients are reported in Table 1.

The median age at enrollment was 53 years (IQR range 44–63 years). The majority of patients (75%) had a PS score of 0, and approximately one-third of the patients were diagnosed with leiomyosarcoma. Other prevalent histologies included liposarcoma, synovial sarcoma, and sarcoma NOS. Regarding tumor grade, 65% of the patients had Grade 3 tumors, 25% had Grade 2, and 6% had Grade 1. For 3 out of 80 patients (4%), information about tumor grade was unknown.

More than half of the patients (61%) had received prior treatments. Surgical intervention was the most common previous treatment modality received by all the patients. Additionally, 34 out of 49 patients (69%) had undergone radiotherapy, and 18 out of 49 patients (37%) had received chemotherapy. Among these, 15 patients received a combination of RT and CT (Table 2).

Out of the total 80 patients, 73 of them (91%) presented with distant metastasis, while the remaining 7 patients had locally advanced malignancy not amenable to surgery. When considering the specific sites and characteristics of distant metastasis, the lung was the most affected site, observed in 73% of the patients. Other sites involved (with or without concomitant lung involvement) were the liver, lymph nodes, abdomen, bone, peritoneum, soft tissue, pleura, pancreas, and heart (Table 3 and Table 4).

Regarding response assessment, all patients enrolled in the study received at least one cycle of treatment with NPLD and ifosfamide. The median number of cycles administered was four, ranging from one to six cycles. As indicated in Table 5, complete response (CR) and partial response (PR) were obtained in 3 patients (4%; 95% CI: 0.00–0.08) and 29 patients (36%; 95% CI: 0.26–0.47), respectively. Additionally, stable disease (SD) was observed in 41% (95% CI: 0.30–0.50) of the patients. The ORR was 40% (95% CI: 0.29–0.51), while the DCR determined as the sum of CR, PR, and SD was achieved in 65 patients (81%; 95% CI: 0.73–0.90). DCR was maintained in 34 patients at 6 months. Progressive disease (PD) was observed in only 16% (95% CI: 0.08–0.24) of the subjects. Two patients who experienced treatment-related toxicity and received only one cycle of chemotherapy were not evaluable. The median DoR was 4 months, with the longest recorded response reaching 152 months.

For survival analysis, the median follow-up was 18.8 months. At the data cut-off point, a total of nine patients were alive, six of whom had no evidence of disease following surgery or local treatments performed after the NPLD and ifosfamide chemotherapy regimen.

Figure 1A reports the Kaplan–Meier curves relative to PFS. The median PFS was 7.9 months (95% CI 5.3–10.4). The percentage of patients with PFS at 1 year was 30%, and at 2 years, it was 13%. The median OS was 21.2 months (95% CI: 12.8–45.9). The percentage of patients with OS at 1 year was 76%; at 2 years, it was 46%; at 3 years, it was 32% and at 5 years, 11% (Figure 2A). For both PFS and OS, the long-rank test was performed considering the different variables at baseline (i.e., sex, age, PS, histology, tumor site, tumor size, grade, state of surgical margins, presence and type/site of metastasis, previous therapy). Interestingly, statistically significant differences were observed in both parameters for STS patients who underwent surgical procedures in combination with RT or CT. Indeed, for PFS (Figure 1B), the median PFS was 7.8 months (95% CI: 4.9–10.4) for patients receiving surgery and CT/RT, compared to 21.1 months (95% CI: 6.2-not evaluable for censored cases) for patients receiving surgery only. Similar trends were observed for OS (Figure 2B) with median values of 23.4 months (95% CI: 15.5–32.6) for patients receiving surgery and CT/RT, and 47.5 months (95% CI 14.0-not evaluable for censored cases) for patients undergoing surgery alone. Upon analyzing the treatments associated with surgery separately, a statistically significant worse outcome was found for patients who had undergone previous RT, while the use of previous CT did not show a statistically significant difference. Although not reaching statistical significance, a better OS and PFS were observed for patients with a better PS (0 compared to 1–2) as well as in patients with a lower tumor grade (low-intermediate compared to high grade).

Regarding toxicity, 44% (95% CI: 0.33–0.55) of patients required a dose reduction. Among them, 29 continued the treatment after reducing the dose to the first level (75%), while 6 patients had to further reduce the dose to the second level (50%).

Thirty-three patients (41%) experienced a delay in treatment administration with a median delay of two weeks. For 61 patients, maintaining the intended dose intensity was not possible, while it was achievable for 17 (21%) patients. Two patients who received only one cycle of therapy were not included in this analysis. Treatment interruption due to toxicity was observed in four patients, with adverse events occurring in two of them after the first cycle of therapy (Table 6).

Overall, grade 4 neutropenia was detected in 30 (38%) patients, while febrile neutropenia, which mostly occurs after the first cycle of treatment and represents the main reason for dose reduction, was observed in 18 (23%) patients. Among the patients who had their dose reduced, grade 4 thrombocytopenia was present in 10, grade 3 thrombocytopenia in 8 patients, while grade 3 anemia was found in 19. In patients who did not undergo a dose reduction, grade 4 thrombocytopenia was observed in four cases, grade 3 thrombocytopenia in one case, and grade 3 anemia in five cases. Other toxicities observed were of grade 1 or 2 severity or, if grade 3, were infrequent (Appendix A).

A reduction in left ventricular ejection fraction (LVEF) of more than 10% was observed in only two patients who had not been previously treated with anthracyclines. Routine echocardiography detected a minor and completely asymptomatic LVEF reduction (ranging from 1% to 7%) in 28 patients (35%). However, from the cardiac evaluation, no cases exhibited heart failure cardiotoxicity (CHF).

After completing the investigated treatment regimen, with the exclusion of five patients, 72 patients received additional therapies. Unfortunately, data regarding the subsequent therapies are either unknown or unavailable for three patients.

The treatment modalities in the study consisted of surgery in 35 cases, RT in 33 cases, and CT in 58 cases. In several cases, as reported in Figure 3A, there was a combination of the different approaches. Among patients who received subsequent CT treatments, the majority (45%) received one additional line of therapy (Figure 3B). As expected, the number of patients receiving further lines decreased progressively, and only one patient received a fifth line of CT. The regimens used in these subsequent chemotherapies are depicted in Figure 3C. Trabectedin emerged as the most frequently used drug (45% of patients), followed by ifosfamide (24%) and gemcitabine either alone or in combination (14% overall). Other drugs were used in isolated cases.

## 4. Discussion

The combination of doxorubicin and ifosfamide is a well-established standard treatment approach for advanced/metastatic STS. However, its effectiveness can vary significantly depending on several factors, including the specific subtype of STS, disease stage, tumor location, and individual patient characteristics. A large randomized controlled phase III EORTC study involving 455 patients with metastatic STS compared doxorubicin monotherapy to a combination of doxorubicin and ifosfamide as first-line therapy. The combination therapy yielded an ORR of 26%, with approximately 77% of patients deriving a benefit from the chemotherapy combination [23]. In a broader context, clinical trials have consistently reported an ORR for the combination therapy ranging from approximately 14% to 34%. However, it is important to note that these outcomes have been accompanied by notable toxicity [26]. Hence, it is crucial to focus on improving both the ORR and mitigating the associated toxicity in order to enhance patient outcomes.

The present study aimed at evaluating the activity and safety of NPLD (50 mg/m^2^ on day 1) in combination with ifosfamide (3000 mg/m^2^ on day 1, 2, and 3 with equidose MESNA) administered every 3 weeks as a first-line treatment for advanced/metastatic STS. The primary endpoint of the study was to assess the ORR, which was achieved with a significant result of 40%, comprising a 36% PR rate and a 4% CR rate. Furthermore, when including SD in the analysis, the study demonstrated a DCR of 81%. Interestingly, the responses were rapid and predominantly detected at the first radiological evaluation, which was planned after two cycles of CT as per protocol.

In another phase II trial involving 34 patients with metastatic or relapsed STS within 6 months of adjuvant treatment, the administration of chemotherapy consisting of 40 mg/m^2^ of NPLD on day 1 and 3000 mg/m^2^ of ifosfamide on days 1–3 achieved an ORR of only 17.7%, with a clinical benefit observed in 55.9% of the patients [27]. Although a subset of 35 individuals in our trial underwent a dose reduction, it is worth considering that the higher dose of NPLD used in our study may have contributed to a more significant tumor response. This suggests that the dosage of NPLD could be a crucial factor in optimizing treatment outcomes for patients with advanced/metastatic STS. However, the treatment outcomes observed in our series could be influenced by the patient population and/or disease characteristics. Among the analyzed patients, only 7 had locally advanced disease, and 12 underwent surgery alone without prior additional treatment. Additionally, 32% of the patients had a histotype compatible with leiomyosarcoma, which is known to be less responsive to the combination of doxorubicin and ifosfamide. Nevertheless, it is noteworthy that within this histological subgroup, a notable 35% achieved a PR in contrast to the findings of the retrospective study conducted by D’Ambrosio et al. [28], where the combination of doxorubicin and ifosfamide yielded a PR rate of 19.5%.

When considering the secondary endpoints, the study revealed a PFS of 7.9 months and an OS of 21.2 months in the entire study population. These values, along with the overall outcome, are higher compared to those obtained in a previous phase II trial using NPLD [27], which reported PFS and OS of 4.2 and 12.7 months, respectively.

Multivariate analysis was conducted to assess the impact of various factors on patient outcomes. The results indicated that patients who underwent surgery in combination with RT and/or CT had a higher risk of death compared to those who received surgery alone. This finding likely reflects the fact that patients selected for surgery and CT or RT are those considered to be at high risk with a worse prognosis. Notably, other variables previously identified as prognostic factors [29,30,31] did not show statistical significance in our study. While we cannot definitively explain this finding, it is possible that the limited number of patients in our study may have influenced the results. The smaller sample size can impact statistical power, making it challenging to detect significant associations between certain variables and prognosis. Further research with a larger cohort is necessary to comprehensively explore these potential prognostic factors. The treatment regimen investigated in this study was generally well tolerated, although hematological toxicities were the most frequently observed adverse events. Grade 3 and 4 neutropenia, in particular, were recorded in 74% of patients. Interestingly, the incidence of febrile neutropenia was remarkably low, detected in only 23% of cases. This is a significant result when compared to the findings of a large phase III trial that assessed the safety of doxorubicin plus ifosfamide versus doxorubicin monotherapy [23]. In that study, the combination arm reported a much higher incidence of febrile neutropenia, as high as 46%, highlighting the favorable outcomes observed in the present trial. Notably, the combination of NPLD and ifosfamide did not result in cardiac toxicity, as only 3% of the subjects experienced a reduction in LVEF greater or equal to 10%. Interestingly, these changes were detected in completely asymptomatic patients during echocardiography performed according to the protocol. Instead, no cases of CHF were detected at the cardiological evaluation. These findings align with the generally observed lower cardiotoxicity of NPLD compared to doxorubicin, which has also been observed in other tumor types, including breast cancer [22,27,32,33].

This study has several limitations. Firstly, the relatively low number of patients may affect the evaluation of different variables, as their impact may be masked by the low frequency. Additionally, the heterogeneity of STS can influence both response and outcome. Finally, being a single-arm study, it only allows for indirect comparison with previously published data.

## 5. Conclusions

In conclusion, the findings of the present study highlight that the combination of NPLD and ifosfamide is an effective treatment option for patients with advanced/metastatic STS, resulting in a satisfactory DCR while maintaining manageable levels of toxicity. The promising outcomes observed in this study underscore the need for further investigation and clinical utilization of NPLD in the treatment of this challenging disease. By continuing to explore the potential of NPLD, we can strive towards improving treatment outcomes and enhancing the quality of life for patients with advanced/metastatic STS.

## Figures and Tables

**Figure 1 cancers-15-05036-f001:**
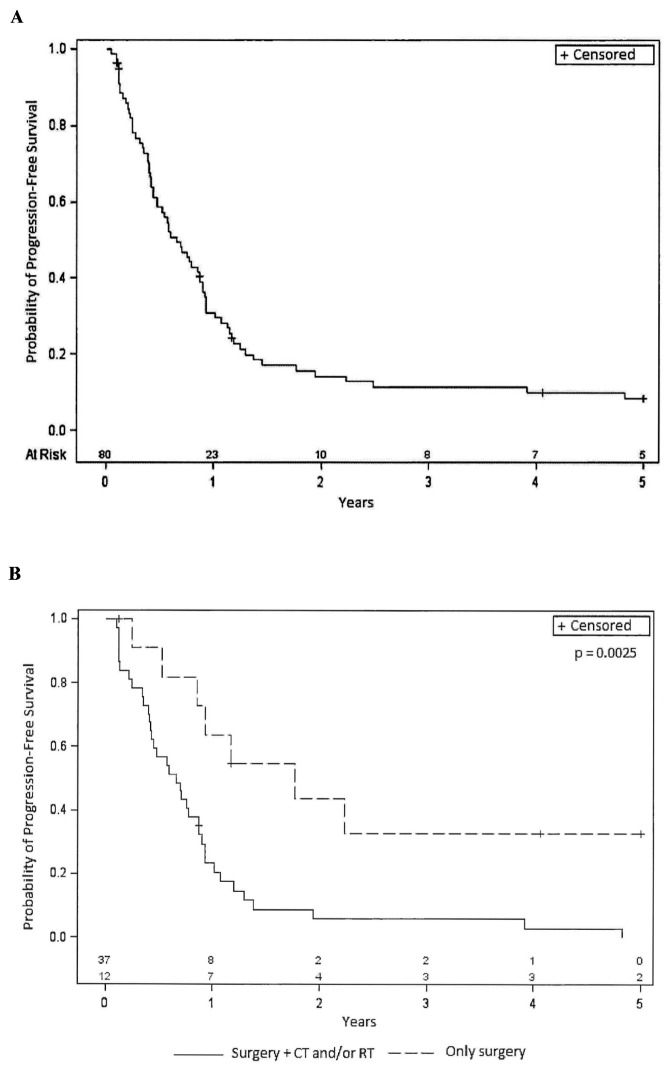
(**A**): Kaplan–Meier curve relative to Progression-Free Survival in the entire population (80 patients). (**B**): Kaplan–Meier curves relative to Progression-Free Survival for patients who underwent previous surgery alone or in combination with other therapies. CT = chemotherapy, RT = radiotherapy, + = censored.

**Figure 2 cancers-15-05036-f002:**
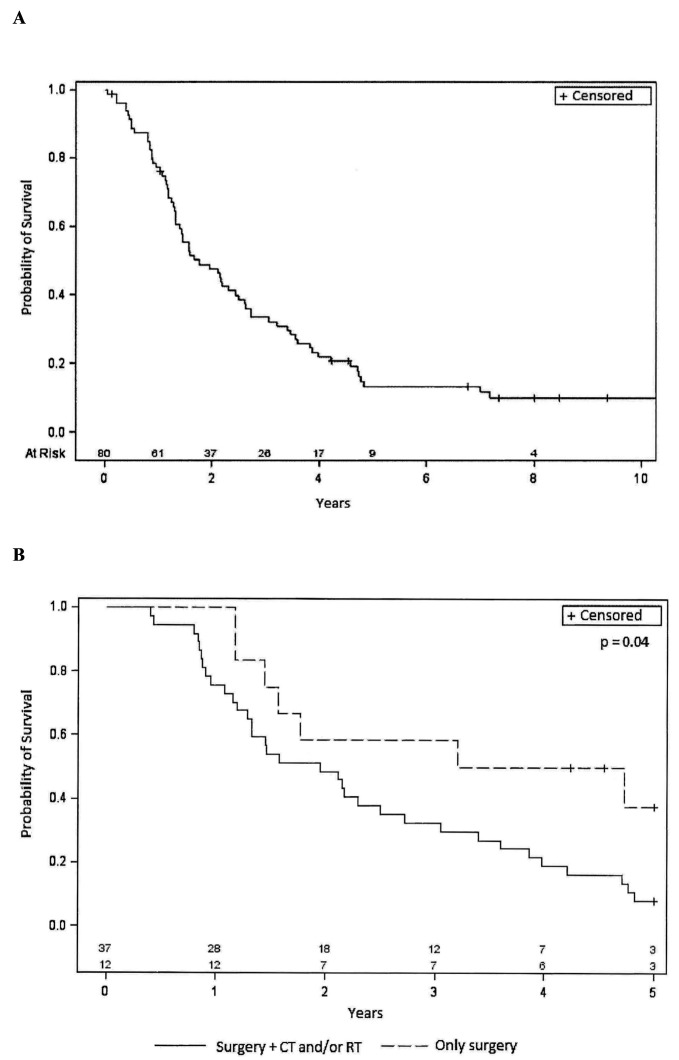
(**A**): Kaplan–Meier curve relative to Overall Survival in the entire population (80 patients). (**B**): Kaplan–Meier curves relative to Overall Survival for patients who underwent previous surgery alone or in combination with other therapies. CT = chemotherapy, RT = radiotherapy, + = censored.

**Figure 3 cancers-15-05036-f003:**
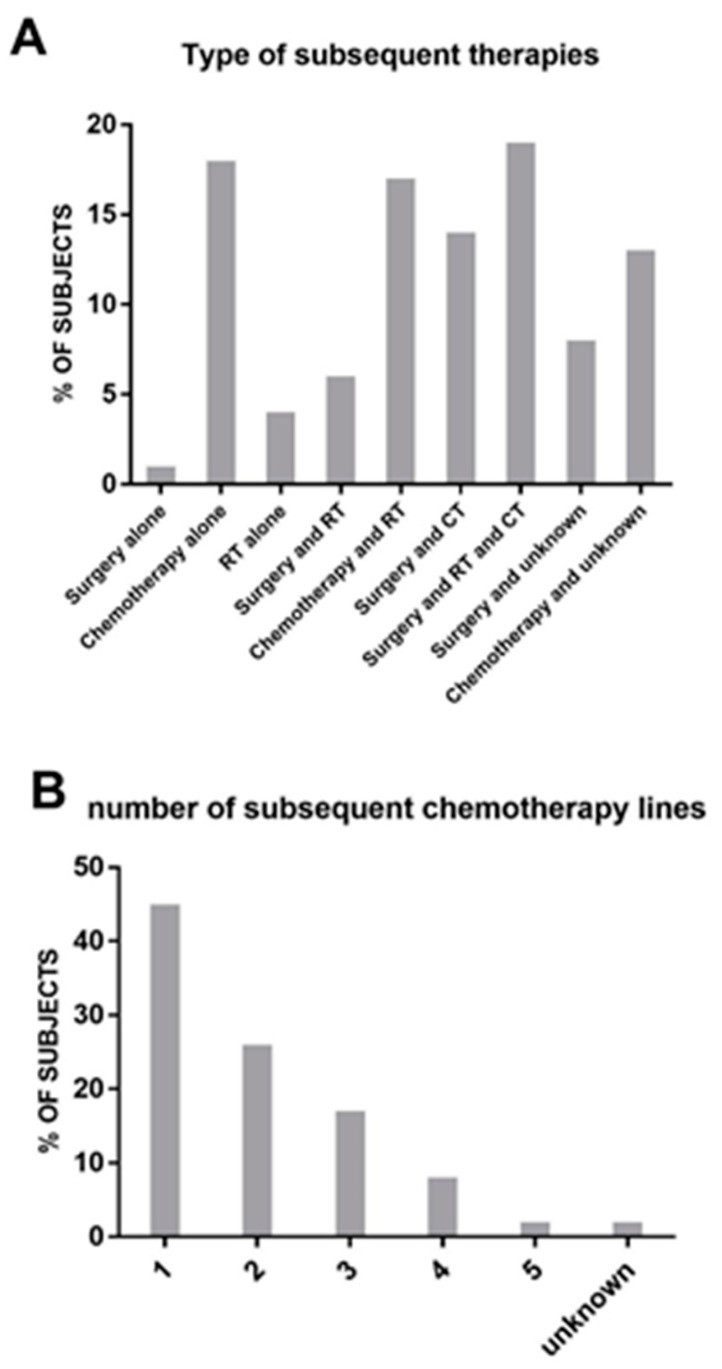
(**A**): List and frequency (in percentage) of therapies received post-protocol. (**B**): Number of subsequent lines of chemotherapy received post-protocol. (**C**): Type of drug used in second-line chemotherapy.

**Table 1 cancers-15-05036-t001:** Clinical characteristics of the patients enrolled in the study.

		N	%
Sex			
	Male	38	47
	Female	42	53
Performance status			
	0	60	75
	1	16	20
	2	4	5
Histotype			
	Leiomyosarcoma	26	32
	Liposarcoma	12	15
	Synovial Sarcoma	11	14
	Sarcoma NOS	11	14
	Angiosarcoma	6	8
	Undifferentiated Pleomorphic Sarcoma	4	5
	MPNST	4	5
	Clear Cell Sarcoma	2	3
	Chondrosarcoma	1	1
	Epithelioid Sarcoma	1	1
	Glomangiosarcoma	1	1
	PNET	1	1
Site of primary tumor			
	Extremity	33	41
	Retroperitoneum	16	20
	Viscera	16	20
	Superficial trunk/abdomen	14	18
	Unknown	1	1
Grading			
	G1	5	6
	G2	20	25
	G3	52	65
	Unknown	3	4
Previous therapies			
	Yes	49	61
	No	17	21
	Unknown	14	18
Presence of distant metastasis			
	Yes	73	91
	No	7	9

**Table 2 cancers-15-05036-t002:** Previous therapy, surgical margins, and local relapse (49 patients).

			N	%
Type of previous therapy				
Surgery			49	100
RT			34	69
CT			18	37
	Surgery alone		12	24
	Surgery and RT		19	39
		Pre-operative RT	13	68
		Post-operative RT	6	32
	Surgery and CT (adjuvant)		3	6
	Surgery and CT/RT		15	31
		Pre-operative CT/RT	9	60
		Post-operative CT/RT	3	20
		Pre-operative RT and perioperative CT	1	7
		Pre-operative CT and perioperative RT	2	13
IORT				
	Yes		11	22
	No		38	78
Surgical margins				
	R0		19	39
	R1		14	29
	R2		3	6
	Unknown		13	26
Local relapse				
	Yes		17	35
	No		32	65

RT, radiotherapy; CT, chemotherapy; IORT, intraoperative radiotherapy.

**Table 3 cancers-15-05036-t003:** Characteristics of distant metastasis involvement (73 patients).

		N	%
Time of onset			
	Synchronous	15	21
	Metachronous	46	63
	Unknown	12	16
Involvement of lung			
	Yes	53	73
	No	19	26
	Unknown	1	1
Site of metastasis			
	Only lung	30	41
	Only extra-lung	19	26
	Lung and extra-lung	23	32
	Unknown	1	1

**Table 4 cancers-15-05036-t004:** Extra-lung involvement (42 patients).

		N	%
Distant tissue involved			
	Liver	15	36
	Lymph nodes	11	26
	Intra-abdominal	11	26
	Bone	8	19
	Peritoneum	6	14
	Soft tissues	5	12
	Pleura	5	12
	Pancreas	1	2
	Heart	1	2

Multiple associations are possible.

**Table 5 cancers-15-05036-t005:** Best response to chemotherapy (80 patients).

		N	%
Best response			
	CR	3	4
	PR	29	36
	SD	33	41
	PD	13	16
	Not evaluable	2	3
ORR		32	40
DCR		65	81

CR, complete response; PR, partial response; SD, stable disease; PD, progressive disease; ORR, overall response rate; DCR, disease control rate.

**Table 6 cancers-15-05036-t006:** Dose reductions, dose delays, and relative dose intensity in patients.

			N	%
Dose reduction				
	No		45	56
	Yes		35	44
		1 level	29	83
		2 levels	6	17
Delayed administration				
	No		45	56
	Yes		33	41
	Not evaluable		2	3
Maintenance of dose intensity				
	Yes		17	21
	No		61	76
	Not evaluable		2	3
Interruption due to toxicity			4	5

## Data Availability

All data can be found in the text.

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
