# Peer review of "Prospective, Multicenter Phase II Trial of Non-Pegylated Liposomal Doxorubicin Combined with Ifosfamide in First-Line Treatment of Advanced/Metastatic Soft Tissue Sarcomas"

_cancers, 2023, doi:10.3390/cancers15205036_

Round 1

Reviewer 1 Report

Typo in line 45

I found this an interesting study, well presented with a strong rationale in terms of maximising response rate and reducing toxicity. The authors present impressive results in comparison to other studies despite including histiotypes that are traditionally viewed as chemo-resistant. However, a significant number of patients were chemo-naive despite having advanced disease which may have impacted the results. I see patients were excluded if PD after 1 cycle - how was this assessed as sometimes we see this after one cycle but disease subsequently stabilises. 

A randomisation between dox and non-pegylated would have been ideal.

It would have been helpful to understand the causes of treatment delay and dose reductions as this was required by a large number of patients. 

Author Response

Point by point answers of the authors to the reviewers

Reviewer 1

I found this an interesting study, well presented with a strong rationale in terms of maximising response rate and reducing toxicity. The authors present impressive results in comparison to other studies despite including histiotypes that are traditionally viewed as chemo-resistant. However, a significant number of patients were chemo-naive despite having advanced disease which may have impacted the results. I see patients were excluded if PD after 1 cycle - how was this assessed as sometimes we see this after one cycle but disease subsequently stabilises. 

Reply: Thank you for your thoughtful and positive feedback on our study. We are delighted to hear that you found our research interesting and well-presented. Your recognition of the strong rationale behind our approach to maximize response rates while reducing toxicity is greatly appreciated.

Although a significant portion of our patients were chemo-naïve, we retain that the study’s outcomes may not have been significantly impacted by this factor. Our study had comparable enrolment criteria of other studies since and in particular mirrored that of Judson et al., in which both chemo-naive and pre-treated patients were enrolled.

We apologize for the lack of clarity in the study design. Indeed, patients whose clinical conditions, deteriorated after first cycle of treatment making their chemotherapeutic journey unfeasible were considered not evaluable. This aspect has been better clarified in the revised version of the manuscript “with the first instrumental evaluation scheduled after the second cycle. Only patients with a stable disease or partial/complete response after the first instrumental evaluation continued the treatment until the subsequent evaluations scheduled after the fourth and sixth cycles of chemotherapy, or until disease progression or unacceptable toxicity. Conversely, patients who experienced clinical deterioration within 3 weeks from starting treatment were deemed not evaluable and excluded from the study”. Line 171-177

A randomisation between dox and non-pegylated would have been ideal.

Reply: We are perfectly in agree with the reviewer, however, in the context of the phase II study the randomization of patients does not allow a comparison between them. Therefore, considering that the expected response rate for first-line treatment with epirubicin and ifosfamide is around 26% (Judson et al), a single-arm study with a sample size of 80 patients was preferred to provide greater statistical power.

It would have been helpful to understand the causes of treatment delay and dose reductions as this was required by a large number of patients. 

Reply: Following the reviewer's advice, we have better explained the reasons behind the dose reductions and delays also by introducing two new supplemental tables which report the rate of toxicity: “Overall, grade 4 neutropenia was detected in 30 (38%) patients, while febrile neutropenia, which mostly occurs after the first cycle of treatment and represents the main reason for dose reduction, was observed in 18 (23%) patients. Among the patients who had their dose reduced, grade 4 thrombocytopenia was present in 10 (8%), grade 3 thrombocytopenia in 8 patients while grade 3 anemia was found in 19 (24%). In patients who did not undergo a dose reduction, grade 4 thrombocytopenia was observed in 4 cases, grade 3 thrombocytopenia in one case, and grade 3 anemia in 5 cases. Other toxicities observed were of grade 1 or 2 severity or, whether of grade 3, infrequent (Table 1S and Table 2S).” Line 334 to 349

Typo in line 45

Reply: As suggest by the reviewer we corrected the type error

Reviewer 2 Report

Systemic treatments of STSs are mainly based on anthracyclines (e.g., doxorubicin) alone or in combination with an alkylating agent (e.g., ifosfamide). This study demonstrates the benefit of the combination of NPLD plus ifosfamide versus PLD plus ifosfamide for advanced/metastatic  STS. Chemotherapy dosage seems to be an important point leading to an optimized ORR compared to lower dose, even if the dose intensity is maintained in only 21% of patients.

Study Design : Could autors explain why "patients who progresseed after completing at least 2 full cycles of treatment were excluded" ? 

Considering the low incidence of this pathology, it would be interesting to study the economic impact of these liposomal forms compared to the classic form. 

Author Response

Point by point answers of the authors to the reviewers

Reviewer 2

Study Design: Could authors explain why "patients who progressed after completing at least 2 full cycles of treatment were excluded"? 

Reply: Thank you for your insightful comments and observations regarding our study on systemic treatments for soft tissue sarcomas (STSs). Your comment about chemotherapy dosage and its impact on the optimized objective response rate (ORR) is crucial, particularly in light of the relatively low percentage of patients who were able to maintain dose intensity.

We apologize for the lack of clarity in the study design. Patients who progressed after two treatment cycles were included in the assessment of ORR but did not continue with the same therapy since they moved to a second-line therapy. This concept is better expressed in the new version of the manuscript as follow: “with the first instrumental evaluation scheduled after the second cycle. Only patients with a stable disease or partial/complete response after the first instrumental evaluation continued the treatment until le subsequent evaluations scheduled after the fourth and sixth cycles of chemotherapy, or until disease progression or unacceptable toxicity. Instead, patients who experienced clinical deterioration within 3 weeks from starting treatment were deemed not evaluable and excluded from the study”. Line 171-177.

Considering the low incidence of this pathology, it would be interesting to study the economic impact of these liposomal forms compared to the classic form. 

Reply: We agree with the reviewer that it would be interesting to study the economic impact of NPLD form compared to the classic form. However, economic issue cannot be covered by this investigation since the lack of a control arm with free doxorubicin which would allow the effective cost comparison of the two treatments including beside the cost of the drugs and that of hospitalization and the support therapies.